# The Effects of T-2 Toxin, Deoxynivalenol, and Fumonisin B1 on Oxidative Stress-Related Genes in the Kidneys of Laying Hens

**DOI:** 10.3390/toxins16030154

**Published:** 2024-03-16

**Authors:** Benjamin Kövesi, Szabina Kulcsár, Zsolt Ancsin, Márta Erdélyi, Erika Zándoki, Patrik Gömbös, Krisztián Balogh, Miklós Mézes

**Affiliations:** 1Department of Feed Safety, Institute of Physiology and Nutrition, Hungarian University of Agriculture and Life Sciences, Szent István Campus, H-2100 Gödöllő, Hungaryballane.erdelyi.marta@uni-mate.hu (M.E.);; 2HUN-REN-MATE Mycotoxins in the Food Chain Research Group, H-7400 Kaposvár, Hungary; 3Agrobiotechnology and Precision Breeding for Food Security National Laboratory, Department of Physiology and Animal Health, Institute of Physiology and Nutrition, Hungarian University of Agriculture and Life Sciences, H-7400 Kaposvár, Hungary

**Keywords:** T-2 toxin, deoxynivalenol, fumonisin B1, laying hens, multi-mycotoxin exposure, oxidative stress, aryl hydrocarbon receptor (Ahr), nuclear factor erythroid-derived 2-like 2 (Nrf2)

## Abstract

In the context of nephrotoxic risks associated with environmental contaminants, this study focused on the impact of mycotoxin exposure on the renal health of laying hens, with particular attention to oxidative stress pathways. Sixty laying hens were assigned to three groups—a control group (CON), a low-dose mycotoxin group (LOW), and a high-dose mycotoxin group (HIGH)—and monitored for 72 h. Mycotoxin contamination involved T-2/HT-2 toxin, DON/3-AcDON/15-AcDON, and FB1 at their EU-recommended levels (low mix) and at double doses (high mix). Clinical assessments revealed no signs of toxicity or notable weight changes. Analysis of the glutathione redox system parameters demonstrated that the reduced glutathione content was lower than that in the controls at 48 h and higher at 72 h. Glutathione peroxidase activity increased in response to mycotoxin exposure. In addition, the gene expression patterns of key redox-sensitive pathways, including Keap1-Nrf2-ARE and the AhR pathway, were examined. Notably, gene expression profiles revealed dynamic responses to mycotoxin exposure over time, underscoring the intricate interplay of redox-related mechanisms in the kidney. This study sheds light on the early effects of mycotoxin mixtures on laying hens’ kidneys and their potential for oxidative stress.

## 1. Introduction

The kidney’s susceptibility to nephrotoxic insults is exacerbated by its robust blood flow and specialized metabolism [1]. Environmental contaminants that target the kidney include metals, solvents, and naturally occurring compounds such as mycotoxins.

Mycotoxins of the *Fusarium* molds, such as trichothecenes (e.g., T-2 toxin, DON) and fumonisins (FBs) can cause acute or chronic toxicity in humans and livestock by contaminating food and feed, often resulting in hepatotoxic, nephrotoxic, and immunotoxic effects [2]. T-2 toxin and deoxynivalenol (DON) are widespread and highly toxic mycotoxins [3]. T-2 toxin exerts a wide range of effects in poultry, including cytotoxicity, genotoxicity, modulation of metabolism, immunotoxicity, hepatotoxicity, and even nephrotoxicity [4]. Exposure to T-2 toxin (0.5 mg/kg feed) induces pathological alterations in the kidneys of poultry, characterized by an increase in relative kidney weight, elevated blood urea nitrogen (BUN) levels, and the presence of vacuolar degeneration in the tubular epithelium, often accompanied by pyknotic nuclei. These observations suggest impaired renal function and have been documented in ducklings and chicks [5,6]. In addition, sporadic cases of renal necrosis have been reported [7]. These pathological changes, and the resulting renal dysfunction, are probably due to the oxidative stress induced by T-2 exposure, which was proven by increased malondialdehyde (MDA) levels at high contamination levels (3.09 mg/kg feed) of T-2 toxin [8]. Liang et al. [9] found that intraperitoneal injection of DON (1.25 mg/kg bw) could significantly increase the oxidative stress-induced apoptosis rate of renal cells in mice. Additionally, DON was found to cause renal dysfunction and oxidative stress in mice by reducing superoxide dismutase activity and inhibiting hydroxyl free radicals. Lei et al. [10] also showed in vitro (porcine kidney cells PK-15) that DON had a more significant toxic effect on porcine kidneys compared to fumonisin B1 (FB1). Szabó et al. [11] also showed a nephrotoxic effect of DON (daily oral administration by gavage; 15 μg), as evidenced by increased MDA in rat kidneys. FB1 presents a different scenario [12]. Avian species are known for their relative resistance to fumonisins [13]. However, recent studies have shown a prolonged persistence of fumonisins in the livers of avian species, which means that fumonisin accumulates in the body [14,15]. It also suggests that the kidneys are less sensitive to fumonisins than the liver in avian species [15]. The cause of the organ-specific sensitivity of FB1 is the different organ-specific accumulation capacity of FB1 [16].

Numerous studies have demonstrated the involvement of oxidative stress in the toxicity of trichothecene mycotoxins and fumonisins commonly found in poultry feed [12,17]. However, there is little to no information on the effects of trichothecenes (DON and its active metabolites, and T-2 and its active metabolite, HT-2 toxin) and FB1, particularly not on their effects in combination. As feeds are often contaminated with different mycotoxins [18], it is critical to understand their collective effects on poultry health.

Oxidative stress is a known effect of most mycotoxins and is the main cause of adverse effects [19,20]. Combined exposure of *Fusarium* mycotoxins caused oxidative stress in vitro in Caco-2 cell lines [21], and in vivo in mice [22]. In both in vitro and in vivo models, a synergistic or additive effect existed between DON and FB1. The cellular defense mechanism against oxidative damage is regulated by the aryl hydrocarbon receptor (AhR) and nuclear factor erythroid-derived 2-like 2 (Nrf2) signaling pathways [23]. The Nrf2 pathway activates as an effect of redox changes in the cells. As a transcription factor regulates the transcription of antioxidant genes [24], the AhR pathway activates the detoxification of xenobiotics, such as mycotoxins [25]. Without xenobiotic exposure, AhR is sequestered by a complex consisting of heat shock protein 90 (Hsp90), hepatitis B virus X-associated protein (XAP2), protein p23, and c-Src (proto-oncogene tyrosine-protein kinase Src) [26]. The activation of the AhR signaling pathway by mycotoxin exposure is due to the act of mycotoxins as AhR ligands. The following signal transmission step is the dissociation of AhR from the above-mentioned protein complex and translocation to the nucleus as ligand-AhR [27]. After the heterodimerization of the ligand-AhR complex with ARNT, it binds to xenobiotic-responsive elements (XREs) and activates the expression of genes encoding the xenobiotic detoxifying enzymes, such as CYP1A1 and CYP1B1 [28].

As mentioned above, the protection against oxidative stress is regulated by both the AhR and Nrf2 signaling pathways [23]. Still, in the case of mycotoxins, the activation of those pathways showed dose dependency. At relatively low doses, the upregulation of CYP450 genes was found. However, downregulation or no effect was found with higher doses of mycotoxins, such as T-2 toxin and DON. The effects of CYP450 and other phase I xenobiotic transforming enzymes play an important role in detoxifying the mycotoxins, resulting in fewer or even more toxic metabolites. Those mycotoxins, which are non-detoxified, and the toxic metabolites induced expression changes in the proteome; therefore, they activated the generation of reactive oxygen species (ROS) and, consequently, oxidative stress [29,30]. Therefore, the primary objective of this study was to assess the early dose-dependent effects of a mixture of three *Fusarium* mycotoxins on the amount of reduced glutathione and activity of glutathione peroxidase, as well as the expression of genes of the redox-sensitive pathways, with a specific focus on the antioxidant response-regulating Keap1-Nrf2-ARE and xenobiotic transformation AhR pathways, in kidney samples obtained from laying hens. This study was conducted over 72 h to comprehensively evaluate the potential oxidative stress and early redox-related effects of mycotoxin exposure in this avian model.

## 2. Results

A short-term (72 h) feeding trial was performed with sixty laying hens. The diets of the experimental groups were contaminated with a mixture of *Fusarium* mycotoxins at low (0.24 mg/kg T-2 and HT-2 toxin: 1.25 mg/kg DON and its acetylated metabolites (3-AcDON and 15-AcDON) and 20 mg/kg FB1) or high (0.46 mg/kg T-2 and HT-2 toxin, 3.65 mg/kg DON and its acetylated metabolites (3-AcDON/15-AcDON), and 40.3 mg/kg FB1) doses.

### 2.1. Clinical Observations, Body Weight, and Relative Kidney Weight

There were no toxic symptoms or mortality observed in the experimental groups. Body and relative kidney weights did not show significant differences between the experimental groups.

### 2.2. Glutathione Content and Glutathione Peroxidase Activity

Reduced glutathione (GSH) concentration was significantly lower as an effect of both multi-mycotoxin doses on day 2, but significantly higher in the low-dose mixture group on day 3 than in the control. Glutathione peroxidase (GPx) activity was significantly higher than in the control for both doses at 48 h, and in the low-dose mixture at 72 h of exposure (Figure 1).

### 2.3. Relative Expression Levels of GPX3, GPX4, GS, and GR Genes

The investigation into the relative expression levels of the glutathione redox system genes—*GPX3*, *GPX4*, *GS*, and *GR*—revealed intriguing patterns (Figure 2). At 24 h, the low-mix dose treatment resulted in a noteworthy upregulation of *GPX3* expression. This trend persisted with both doses on day 3. However, a contrasting downregulation was observed on day 2 following high-dose treatment compared to the control group.

In the case of *GPX4*, a significant downregulation was evident on day 1 for both low and high doses. Nevertheless, a subsequent reversal occurred, with upregulation noted on both day 2 and day 3 under the influences of both doses.

*GS* expression exhibited distinct dynamics, with a significant decrease detected on day 1 across both doses. Conversely, on days 2 and 3, a remarkable elevation in expression was observed, demonstrating a dose-dependent modulation among the treatment groups.

Finally, *GR* displayed differential expression patterns. On day 1, both low and high doses significantly reduced *GR* expression. In contrast, on both day 2 and day 3, *GR* expression was substantially increased within the treatment groups, surpassing that of the control. The treatment and sampling time significantly affected the glutathione redox system encoding genes. A significant treatment × time effect was also found.

### 2.4. Relative Expression Levels of Genes KEAP1 and NRF2

On day 1, *KEAP1* expression was significantly upregulated, but it was followed by a decrease on day 3 in response to the administered doses (Figure 3).

Regarding *NRF2* gene expression, a substantial increase was observed in both dose groups. However, on day 2, expression levels decreased in the high-dose group, which persisted on day 3 for both dose groups (Figure 3). In the cases of *KEAP1* and *NRF2* gene expression, treatment and sampling time had significant effects, and a significant treatment × time effect was found.

### 2.5. Relative Expression Levels of AHR, AHRR, HSP90, and CYP1A2 Genes

On day 1, there was a noteworthy divergence between *AHR* and *AHRR* gene expression levels. Relative expression of *AHR* decreased significantly in both dose groups compared to the control, while *AHRR* exhibited a dose-dependent increase. On day 2, low and high doses resulted in the upregulation of both genes. However, by day 3, *AHR* showed a continued upregulation, whereas *AHRR* experienced downregulation in both dose groups (Figure 4).

The expression pattern of *HSP90* mirrored that of *AHRR*, with dose-dependent overexpression on day 1 persisting on day 2, and then transitioning to downregulation on day 3 for both dose groups (Figure 4).

In the case of *CYP1A2*, a significant upregulation was observed on day 1, followed by downregulation on day 3 in response to both doses (Figure 4). In the cases of both *AHRR* and *HSP90* gene expression levels, treatment and sampling time had significant effects. A significant treatment × time effect was also found. However, in the cases of *AHR* and *CYP1A2,* only time and treatment × time significantly affected gene expression.

## 3. Discussion

Combined mycotoxin exposure is a common problem in feed commodities, due to the presence of different toxigenic molds and their mycotoxins [31]. The proposed maximum levels of mycotoxins in poultry feed in the European Union are 5 mg/kg for DON, 0.25 mg/kg for T-2+HT-2 toxin, and 20 mg/kg for FB1. These proposed levels may apply to individual mycotoxins, but there are no proposals for their presentations in combination, which may modify their toxic effects [32]. Combined exposure to mycotoxins may cause additive, synergistic, or antagonistic interactions [33,34]. These should be considered when determining their toxic effects, which would be important in mycotoxin risk assessment.

There are few in vivo studies investigating the effects of mycotoxins on the expression of AhR signaling pathway genes in the kidneys of poultry species. However, overexpressed AhR and CYP genes were observed in chicken livers due to T-2 toxin [35], and combined exposure to DON and its acetylated metabolites (3-AcDON and 15-Ac-DON), T-2+HT-2 toxin, and FB1 were demonstrated in the livers of laying hens [36]. In this study, the expression levels of the *AHR*, *AHRR*, *HSP90*, and *CYP1A2* genes changed differently at the three sampling times due to *Fusarium* multi-mycotoxin treatment in the kidneys of laying hens. A correlation was found between the expression levels of the *AHR* and *ARHH* genes. Overexpression of *AHRR* and downregulation of *AHR* expression were found on day 1 of the experiment. On day 2, both *AHR* and *AHRR* mRNA levels were higher than those in the control, and on day 3, an opposite tendency to day 1 was found because *AHRR* gene expression was much lower than the *AHR* mRNA level. These results explain that *AHRR* gene expression changed in response to the expression of *AHR* to maintain the physiological level of AhR. Dose-related changes in *AHRR* expression were found only on day 1. This result suggests that the effect of the higher mycotoxin dose was manifested only in *AHRR* gene expression, and only in the early phase of mycotoxin exposure. It was suggested that, in the early phase of mycotoxin exposure, the main route of mycotoxin metabolism at high doses was not through the AhR-activated phase I biotransformation pathway [29]. In the later phase of mycotoxin exposure, the effects of the two mycotoxin doses were the same due to the activation of the AhR signaling pathway. This result suggests that the early response to mycotoxin exposure may cause downregulation of *AHR* gene expression due to high *AHRR* expression. Overexpression of the *AHRR* gene would be a response against high AhR protein expression, keeping it at physiological levels. Dose-dependent overexpression of *HSP90* on day 1 remained high on day 2, but was downregulated on day 3. These results suggest that the early response to the high mycotoxin dose does not activate the canonical AhR, but mainly manifests through the Hsp90 pathway. This dose-dependent pathway of mycotoxin metabolism was hypothesized by Wen et al. [29]. However, the trend of *HSP90* expression was the same as that of *AHRR*, suggesting that the high relative expression of the *HSP90* gene was a response to elevated *AHRR* mRNA levels, since Hsp90 protein in the cytoplasm requires AhR translocation to the nucleus. The changes in gene expression for the AhR pathway were the same as those found in the livers of laying hens [36].

The AhR is a transcription factor that activates the genes encoding the xenobiotic transformation enzymes, such as CYP1A2. The expression of the *CYP1A2* gene showed changes that were opposite to those of *AHR*, but its tendency was the same as those of *HSP90* and *AHRR.* The possible cause of this opposite tendency would be that *CYP1A2* expression was an early response to mycotoxin exposure at both doses. It is suggested that active Phase I biotransformation at a low dose and partial biotransformation at a high dose occurred simultaneously in the cells, as suggested by Wen et al. [29]. In the later phase of mycotoxin exposure (day 3), overexpression of the *AHR* gene was found, but the expression levels of the *AHRR*, *HSP90*, and *CYP1A2* genes were downregulated at this sampling. This means that the effect of the AhR protein manifested later than the end of the experiment and was a response to continuous mycotoxin exposure, regardless of the dose. Otherwise, the CYP1A2 enzyme is an important part of the Phase I xenobiotic transformation, which may lead to more reactive metabolites and, consequently, the generation of ROS [29,30]. In contrast to the kidneys, dose-dependent changes were found in the livers of laying hens. The lower dose revealed a more marked increase than the higher one, and expression of the *CYP1A2* gene was upregulated on days 2 and 3 [36].

Nrf2 is a transcription factor that activates the antioxidant response element (ARE) gene cluster. Genes in this cluster encode molecules that maintain cellular redox homeostasis [37]. In this aspect, there is a correlation between the two transcription factors, Nrf2 and AhR. The Nrf2 protein is sequestered in the cytoplasm by its negative regulator, Kelch-like ECH-associated protein-1 (Keap1), under physiological conditions [38]. Redox changes, and subsequent ROS formation, result in the oxidation of the cysteine sites of Keap1 required for Nrf2 binding. This results in a dissociation of the Keap1-Nrf2 complex, and the Nrf2 binds to ARE in the nucleus, increasing the transcription of genes encoding the Phase II enzymes of xenobiotic transformation [24]. In vivo and in vitro studies have shown that the Nrf2 signaling pathway is impaired by the *Fusarium* mycotoxins [39]. The relative expression levels of the *KEAP1* and *NRF2* genes showed the same trend in the present study. In the case of *KEAP1*, an increase was found on day 1, and the changes depended on the mycotoxin dose applied. The relative gene expression of *NRF2* was high on day 1, and the lower dose showed a higher value. An opposite tendency was found in the livers of laying hens because *NRF2* gene expression showed a dose-dependent downregulation on day 1 [36]. The changes in *NRF2* expression on day 1 were the same as for *GPX3*, but not for *GPX4*, *GS*, and *GR*, mRNA levels. The reason for the difference between *GPX3* and *GPX4* levels on day 1 would be the specific expression of glutathione peroxidase genes. GPx3 mainly originated from renal tubular cells [40], and its expression is the highest in the kidney and lower in other tissues [41], whereas GPx4 has a broad tissue origin, especially in the testis [42]. The results suggest that the mycotoxin-induced redox changes caused by the low dose of the *Fusarium* mycotoxin mixture activate the Nrf2-ARE pathway and, mainly, the gene expression of kidney-specific GPX3. This finding is supported by our previous studies, showing that a combination of DON and T-2 toxin dose-dependently increased *GPX4* expression in chicken livers [43], and a low dose of *Fusarium* multi-mycotoxin exposure elevated the expression of the *GPX4* gene in the livers of laying hens [44]. The relative gene expression of *NRF2* increased at 24 h, and downregulation was found at both 48 and 72 h in the high-mix treatment group. This result suggests that the higher dose of *Fusarium* mycotoxin mixture caused a less effective antioxidant response through this signaling pathway. However, *GPX4*, *GS*, and *GR* gene expression levels did not support this hypothesis, as their gene expression levels were higher on day 2, and *GPX3* gene expression was higher on day 3 than in the control. This phenomenon can be explained by the fact that the *NRF2* protein is phosphorylated before it is transferred to the nucleus, and some additional conformational changes occur. This means that the expression of the *NRF2* gene does not imply the same gene expression changes in the ARE gene cluster. Changes in the relative gene expression levels of the *GS* and *GR* genes correlated with the levels of *GPX3* and *GPX4*. This correlation can be explained by the fact that the enzyme proteins encoded by GS and GR enzymes require the maintenance of optimal levels of the co-substrate of glutathione peroxidases, the reduced glutathione in cells. Glutathione is the most abundant cellular antioxidant [45] and is the co-substrate of glutathione peroxidases [46]. The relative expression levels of the glutathione redox system-related genes showed different changes in the kidneys than in the liver. A more marked upregulation in the kidneys was found at 48 and 72 h, and it was dose-dependent. In this context, the higher dose caused more marked overregulation in the liver than in the kidneys [36]. Changes in the expression of glutathione redox system genes were manifested at glutathione and GPx levels. On day 1, there were no differences; on day 2, glutathione levels were lower and GPx activity was higher than those in the control; on day 3, glutathione levels and GPx activity were higher in the low-dose group than in the control. These results suggested that the higher GPx activity on day 2 caused glutathione oxidation. However, glutathione reduction occurred on day 3 due to GR activation and glutathione synthesis by GS. The hydroperoxide co-substrate causes the discrepancy between *GPX3* gene expression and GPx activity. The biochemical analysis determines mainly GPx4 due to t-butyl hydroperoxide, but the main co-substrate of GPx3 is hydrogen peroxide [47].

## 4. Conclusions

The results of this study suggested that, in the early phase of *Fusarium* multi-mycotoxin exposure, the genes of AhR-activated phase I biotransformation, mainly through Hsp90, and the genes of Nrf2-activated glutathione-dependent antioxidant signaling pathways were activated. The expression levels of several genes, such as *AHRR, HSP90, KEAP1, NRF2, GPX3*, and *GPX4,* revealed marked changes in the early phase of mycotoxin exposure, and these changes were different according to the dose applied. In the later phase of mycotoxin exposure, there were dose-dependent changes in the expression levels of the *HSP90*, *KEAP1*, *NRF2*, and *GS* genes. These changes suggested differences in the activities of the two pathways, which were activated simultaneously.

## 5. Materials and Methods

### 5.1. Birds and Experimental Design

This study used sixty 49-week-old Tetra SL laying hens, randomly allocated to three treatment groups (n = 18). Six animals were selected as absolute controls at the beginning of the experiment.

The experiment started after 12 h of feed deprivation, and mycotoxin exposure occurred over 72 h. Throughout the trial, the hens had ad libitum access to feed and water. They were housed in deep litter. A light schedule of 16 h of light, followed by 8 h of darkness, was maintained during the experiment in accordance with animal welfare guidelines for laying hens [48]. The calculated nutrient content of the diet was as follows: 11.97 MJ/kg ME, 89.20% dry matter, 16.10% crude protein, 2.50% ether extract, 5.50% crude fiber, 0.79% lysine, 0.38% methionine, 0.71% methionine + cysteine, 4.12% calcium, 0.48% available phosphorus, and 0.17% sodium.

Our study involved three distinct treatment groups, as follows:Control (CON): hens received a basal diet.Low-mix Group (LOW): hens received an experimentally contaminated basal diet. The mycotoxin contents were T-2+HT-2 toxin (0.24 mg), DON+ 3-AcDON+15-AcDON (1.25 mg), and FB1 (20 mg/kg feed).High-mix Group (HIGH): hens received an experimentally contaminated basal diet. The mycotoxin contents were T-2+ HT-2 toxin (0.46 mg), DON+3-AcDON+15-AcDON (3.65 mg), and FB1 (40.3 mg/kg feed).

Kidney samples were taken at necropsy from six randomly selected animals within each group at three different sampling points (24, 48, and 72 h). These samples were treated with cold (4 °C) isotonic NaCl solution, flash-frozen in liquid nitrogen, and stored at −80 °C until analysis.

### 5.2. Production and Determination of Mycotoxins

The production of mycotoxins and the contamination of the basal diet were performed according to procedures described in a previous study [44]. The mycotoxins and their toxic metabolites, including T-2and /HT-2 toxin, DON and its acetylated metabolites (3-AcDON and 15-AcDON), and FB1, were quantified in triplicate using a liquid chromatography–mass spectrometry (LC/MS) method, as follows. Mycotoxins of the feed samples were extracted with acetonitrile/water/formic acid (49/49/2 v/v/v), and a mixture of 0.8 g anhydrous MgSO4 and 0.2 g NaCl powder was added and vortexed. The mixture was shaken and centrifuged for 5 min at 4000 rpm. The supernatant acetonitrile phase was diluted to 1 mL with deionized water and filtrated with a 0.22 µm syringe filter. The prepared sample was injected into the LCMS system. Concentrations of mycotoxins in prepared samples were determined with a Shimadzu 2020 LCMS system (Shimadzu, Kyoto, Japan) with an electrospray ion source (ESI). The XB-C18 Kinetex analytical column (100 × 2.1 mm, 2.6 µm; Phenomenex, California, USA) was used with a 0.3 mL/min flow rate and a column temperature of 40 °C. The gradient elution was performed with eluents A (0.1% formic acid + 0.005 M ammonium formate) and B (0.1% formic acid in acetonitrile). The following gradient program was used: the eluent B grade started from 10% and increased linearly in 8 min to 100%; the column was washed with eluent B for 3 min, and the initial conditions were reestablished with decreasing the eluent B linearly in 1 min; then, the column was re-equilibrated for 3 min with 10% eluent B. The measured mycotoxin conents of the feeds is shown in Table 1.

### 5.3. Biochemical Analyses

The concentrations of total non-protein sulfhydryl groups, expressed as GSH [49], and the activity of GPx [50], were measured in the 10,000 g supernatant fraction of the 1:9 kidney homogenate. GSH content and GPx activity were calculated to the protein content of the supernatant fraction using the Folin–Ciocalteu phenol reagent [51].

### 5.4. RNA Extraction and Reverse Transcription

Total RNA was extracted from kidney samples using the NucleoZOL reagent (Macherey-Nagel, Düren, Germany). Genomic DNA contamination of the total RNA was eliminated with DNase I (Thermo Fisher, San Jose, CA, USA). The absorbance ratios at 260 and 280 nm, using a nanophotometer (Implen, Munich, Germany), were used to assess the purity and quantity of the extracted total RNA. Samples with absorbance ratios greater than 2.0 indicated high-quality RNA suitable for cDNA synthesis. The integrity of the RNA samples was further verified by subjecting them to electrophoresis on a 1.5% agarose gel. For reverse transcription, 1 µg of total RNA from each sample was converted to complementary DNA (cDNA) using the High-Capacity cDNA Reverse Transcription Kit according to the manufacturer’s instructions (Thermo Fisher Scientific, San José, CA, USA).

### 5.5. Quantitative Real-Time Polymerase Chain Reaction (qPCR)

The StepOnePlus^TM^ Real-Time PCR System from Applied Biosystems^TM^ was used for qPCR analysis. Reactions were prepared using PowerUp^TM^ SYBR^TM^ Green Master Mix from Thermo Fisher (San José) for qPCR. For each qPCR reaction, we used a 10 µL mixture containing 5 µL PowerUp SYBR Green Master Mix for qPCR, 0.2 µL each of forward and reverse primers (each at 10 µM concentration), 1.6 µL synthesized cDNA, and 3 µL nuclease-free water.

We used a qPCR method known as touchdown qPCR (TqPCR), with slight modifications, based on the protocol described by Zhang et al. [52]. TqPCR reactions were performed in triplicate, according to the following thermal cycling profile: initial denaturation at 95 °C for 10 min, followed by 1 cycle; then, 36 cycles of denaturation at 95 °C for 15 s; annealing at 58 °C for 30 s; and extension at 72 °C for 30 s. In the initial phase, the annealing temperature started at 64 °C and was decreased by 2 °C per cycle.

To validate the specificity of the PCR amplified product, a melting curve analysis was performed. The protocol was as follows: 95 °C for 15 s, 60 °C for 1 min, and 95 °C for 15 s (melt curve step). The method described by Pfaffl was used to determine the relative expression levels of the target genes. [53]. The efficiency (E) of the PCR primer pairs was determined by analyzing the slope of the standard curve, generated through 5-fold serial dilutions of pooled cDNA samples. The efficiency was calculated using the formula E = [10^(−1/slope)^ − 1] × 100%, as recommended by Bustin et al. [54]. All standard curves showed correlation coefficients (r^2^) greater than 0.99, with amplification efficiencies between 90% and 110%. Detailed information on the specific primer sequences used in this study is provided in Table 2.

Melt curve analysis confirmed the presence of a single expected amplification product, as indicated by the melting peaks. In addition, each primer pair produced a single peak on the melting curve and a single band of the expected size with 2% agarose gel electrophoresis.

### 5.6. Statistical Analysis

Data were presented as means and standard deviations (SDs). The Shapiro–Wilk test was used to assess the normality of distribution, and the Bartlett and Brown–Forsythe tests were used to assess the homogeneity of variance. For data sets meeting these criteria, two-way analysis of variance (ANOVA) was performed, using GraphPad PRISM version 9.5.1 software (GraphPad, San Diego, CA, USA).

Differences between means were tested using Tukey’s post hoc test. Statistical significance was considered when *p*-values were less than 0.05.

## Figures and Tables

**Figure 1 toxins-16-00154-f001:**
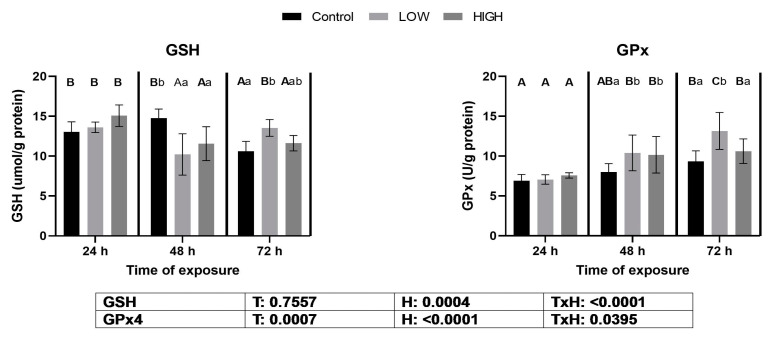
Effects of two multi-mycotoxin doses on reduced glutathione content and glutathione peroxidase activity in the kidneys of laying hens. Ctr, control group. Low mix: T-2+HT-2 toxin: 0.24 mg; DON+3-AcDON+15-AcDON: 1.25 mg; FB1: 20 mg/kg feed. High mix, T-2+HT-2 toxin: 0.46 mg; DON+3-AcDON+15-AcDON: 3.65 mg; FB1: 40.3 mg/kg feed. Data are presented as mean ± SD; n = 6. Lowercase letters indicate significant differences between treatment groups at the same time. Capital letters indicate significant differences between times for the same treatment. T = treatment effect. H = time effect. T × H = treatment × time effect.

**Figure 2 toxins-16-00154-f002:**
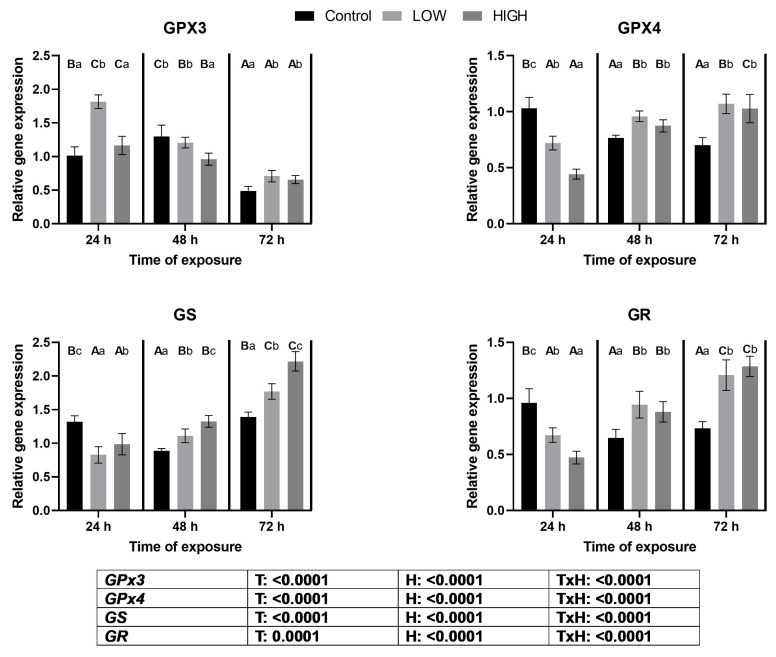
Effects of two multi-mycotoxin doses on the relative expression levels of the *GPX3*, *GPX4*, *GS*, and *GR* genes in the kidneys of laying hens. Ctr, control group. Low mix, T-2+HT-2 toxin: 0.24 mg; DON+3-AcDON+15-AcDON: 1.25 mg; FB1: 20 mg/kg feed. High mix, T-2+HT-2 toxin: 0.46 mg; DON+3-AcDON+15-AcDON: 3.65 mg; FB1: 40.3 mg/kg feed. Data are presented as mean ± SD; n = 6. Lowercase letters indicate significant differences between treatment groups at the same time. Capital letters indicate significant differences between times for the same treatment. T = treatment effect. H = time effect. T × H = treatment × time effect.

**Figure 3 toxins-16-00154-f003:**
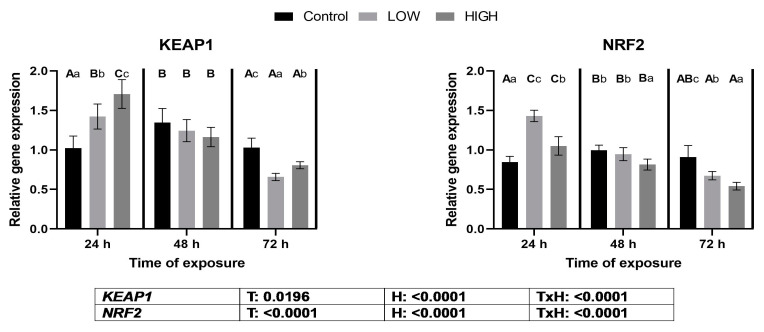
Effects of two multi-mycotoxin doses on the relative expression levels of the *KEAP1* and *NRF2* genes in the kidneys of laying hens. Ctr, control group. Low mix, T-2+HT-2 toxin: 0.24 mg; DON+3-AcDON+15-AcDON: 1.25 mg; FB1: 20 mg/kg feed. High mix, T-2+HT-2 toxin: 0.46 mg; DON+3-AcDON+15-AcDON: 3.65 mg; FB1: 40.3 mg/kg feed. Data are presented as mean ± SD; n = 6. Lowercase letters indicate significant differences between treatment groups at the same time. Capital letters indicate significant differences between times for the same treatment. T = treatment effect. H = time effect. T × H = treatment × time effect.

**Figure 4 toxins-16-00154-f004:**
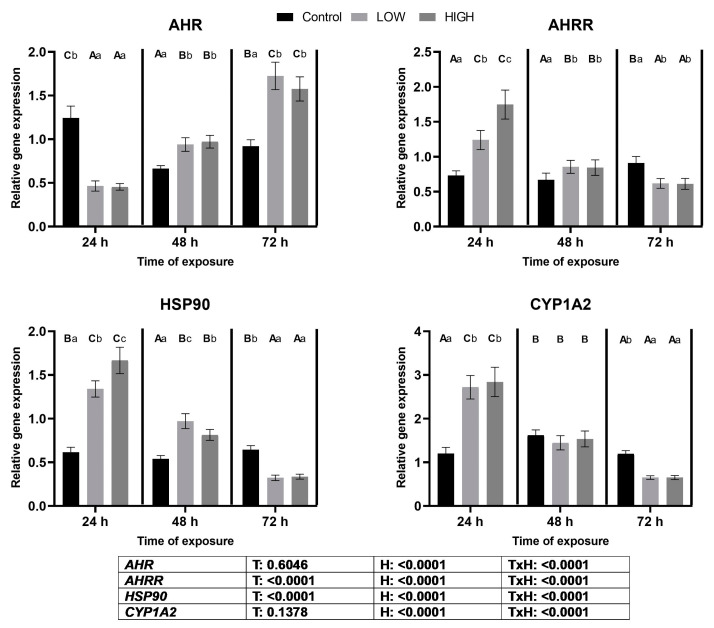
Effect of two multi-mycotoxin doses on the relative expression levels of the *AHR*, *AHRR*, *HSP90*, and *CYP1A2* genes in the kidneys of laying hens. Ctr, control group. Low mix, T-2+HT-2 toxin: 0.24 mg; DON+3-AcDON+15-AcDON: 1.25 mg; FB1: 20 mg/kg feed. High mix, T-2+HT-2 toxin: 0.46 mg; DON+/3-AcDON+15-AcDON: 3.65 mg; FB1: 40.3 mg/kg feed. Data are presented as mean ± SD; n = 6. Lowercase letters indicate significant differences between treatment groups at the same time. Capital letters indicate significant differences between times for the same treatment. T = treatment effect. H = time effect. T × H = treatment × time effect.

**Table 1 toxins-16-00154-t001:** The measured mycotoxin contents of the feeds (mg/kg).

Group	T-2/HT-2	DON/3-AcDON/15-AcDON	FB1
Control	<0.01/˂0.01	<0.02/0.02/<0.02	0.2
Low dose	0.13/0.11	0.67/0.57/0.01	20.0
High dose	0.30/0.16	2.70/0.89/0.06	40.3

**Table 2 toxins-16-00154-t002:** Primer sequences and parameters.

Genes	GenBank Accession No.	Primer Sequences, 5′-3′	Length, bp.	Efficiency, %
Internal controls				
*GAPDH*	NM_204305.1	F-TGACCTGCCGTCTGGAGAAA R-TGTGTATCCTAGGATGCCCTTCAG	98	92.64
*BAC*	NM_205518.2	F-GACGAGATTGGCATGGCTTTATTT R-TAAGACTGCTGCTGACACCTTC	92	96.29
*RPL13*	NM_204999.2	F-GCTTAAACTGGCGGGCATTAAC R-GGCTTGCAGTGACTCTGTAGAT	97	94.97
Target genes				
*KEAP1*	KU321503.1	F-CATCGGCATCGCCAACTT R-TGAAGAACTCCTCCTGCTTGGA	113	99.74
*NRF2*	NM_205117.1	F-TTTTCGCAGAGCACAGATAC R-GGAGAAGCCTCATTGTCATC	110	91.74
*GPX3*	NM_001163232.2	F-ATCCCCTTCCGAAAGTACGC R-GACGACAAGTCCATAGGGCC	129	102.51
*GPX4*	NM_001346448.1	F-AGTGCCATCAAGTGGAACTTCAC R-TTCAAGGCAGGCCGTCAT	203	91.03
*GS*	XM_425692.6	F-GTACTCACTGGATGTGGGTGAAGA R-CGGCTCGATCTTGTCCATCAG	196	104.84
*GR*	XM_015276627.2	F-CCACCAGAAAGGGGATCTACG R-ACAGAGATGGCTTCATCTTCAGTG	208	91.76
*AHR*	NM_204118.3	F-GAAGACGGGTGAGAGTGGAA R-CGCTTCCGTAGATGTTCTGC	171	99.20
*AHRR*	NM_001201387.2	F-AGAACGGCACCATGAGGAAG R-CAGAGGTCCGGTTCTGCTTT	73	99.86
*HSP90*	NM_001109785.2	F-TGAAACACTGAGGCAGAAGG R-AAAGCCAGAGGACAGGAGAG	100	95.05
*CYP1A2*	NM_205146.3	F-ACGCAGATCCCAAACGAGAA R-GTCAAAGCCTGCTCCAAAGATG	63	102.39

## Data Availability

The raw data supporting the conclusions of this manuscript will be made available by the authors, without undue reservation, to any qualified researcher.

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
