# Peer review of "The Effects of T-2 Toxin, Deoxynivalenol, and Fumonisin B1 on Oxidative Stress-Related Genes in the Kidneys of Laying Hens"

_toxins, 2024, doi:10.3390/toxins16030154_

Round 1
Reviewer 1 Report
Comments and Suggestions for Authors
Major comments:
1- This manuscript presents different oxidative stress variables measured at kidney level in laying hens during exposure to a mixture of mycotoxins. The general presentation of the work is very similar to a study carried out by the same authors on the liver (ref 46 of the manuscript). Although this information is not provided, it probably involves some of the same animals. This major point needs to be clarified. It can be accepted, but this should be said. Nevertheless, the originality of the proposed work should better highlight what is expected from the study with kidney, in relation to, and in comparison with, the already published effects measured for the liver.
2- The introduction and discussion should better present what is expected in terms of results and what has been observed. For example, in the introduction (L36-54), the authors mention the kidney toxicity of T2 toxin, but what about its liver toxicity? The comparison of FB1 levels in the liver and kidneys (L56-63) suggests that the renal effects will be less marked than the hepatic effects, and this should be a major point of discussion. If possible, correlations between the effects observed in the liver and kidneys depending on the animal analysed could even be explored.
Moreover, the introduction and the discussion should focus on the mycotoxins administered. By contrast, the various mechanistic hypotheses that could lead to oxidative stress should be considered. Focusing only on the biotransformation systems is not justified as no original result is presented in this way in this study.
3- Concerning the interpretation of the results, part of the effects reported appear to be attributable to variability in the results obtained in the controls depending on the day of exposure (or analysis?). So it is sometimes difficult to draw conclusions about the effects of the toxins. This variability in controls is not explained by the authors. The authors state that they performed a 2-factor ANOVA, and it is assumed that they measured the exposure time effect and the toxin dose effect. This point should be clarified. Full statistical data, including interaction between the variables explored, should be provided. If one of the factors tested or interaction is significant, the comparison of means (Tukey's post hoc test L326) can be carried out, but it should be carried out on the 9 groups studied. It seems that the comparisons were actually made separately on different days, which in my opinion is not acceptable. Analysis on the 9 groups studied would make it possible to rule out differences between controls.
4- The material and methods should be completed. Regarding major comment 1, it could be clarified in relation to reference 46.
Detailed comments
L36-54: focus on trichothecenes, remove anything that is not relevant: ZEN, AFB1. Specify the doses of trichothecenes for which nephrotoxicity has been observed in poultry. What about hepatotoxicity?
L56-64: Specify the doses of fumonisins for which nephrotoxicity has been observed in poultry.
L77-81: As the results section comes before the materials and methods section, one or two brief introductory sentences should be used to explain what was done in terms of groups and doses.
L82-87, Figure 1: the way in which the results are interpreted is difficult to follow. For example, the data in Figure 2 suggest a time effect at 48 and 72 h on GSH concentrations, but this time effect seems to be mainly related to the variability observed in the controls. Provide full statistical data, including interaction analysis.
L95-112, Figure 2: Same remark as for the activities. The variability observed in the controls explains the interpretation of the results, but the results should be interpreted globally. The expression of GPX3 is probably reduced as the dose and duration of exposure increase, whereas the expression of GPX4, GS and GR would be increased.
L119-124, Figure 3: Same comment.
L131-143, Figure 4: Same comment.
L167: provide specific references indicating how the hydrocarbon receptor (AhR) is involved in the detoxification of the mycotoxins administered in this study
L180: exclude results with AFB1, better discuss previous results with fumonisins and trichothecenes. As it is probably the same animals have investigated correlations?
L184-205: this interpretation of the results should be supported by the results of the statistical analyses
L207-265: far too general. If you want to refer to a possible action of biotransformation enzymes, phase 1 and 2, you should focus on the mycotoxins administered. To my knowledge, fumonisins are metabolised only to a limited extent, if at all. As no results on toxins and their metabolites were produced in this study, all this seems to me to be largely speculative. The discussion could also include a discussion on the endoplasmic reticulum stress, which is observed during exposure to trichothecenes and fumonisins, and is responsible for oxidative damages.
L303: reference 46 does not provide this information.
L304-307: It is quite surprising to see different methods from those cited in [46]. The same applies to the levels of toxins in feed, even though the same study is involved.
Why FB1 alone when the recommendations cited concern FB1+FB2?
Reviewer 2 Report
Comments and Suggestions for Authors
The paper titled "Effects of T-2 Toxin, Deoxynivalenol, and Fumonisin B1 on Oxidative Stress-Related Genes in Laying Hen Kidneys" investigates the impact of mycotoxin exposure on renal health in laying hens, focusing on oxidative stress pathways. The study involved three groups of hens exposed to low and high doses of mycotoxins, observing the effects on the glutathione redox system, gene expression related to oxidative stress pathways, and kidney function. The findings reveal dynamic responses in gene expression over time, highlighting the complex interplay of redox-related mechanisms in response to mycotoxin exposure. The study's key contributions include insights into the early effects of mycotoxin mixtures on kidney health and the activation of specific oxidative stress response pathways.
1. Assessment of Individual Toxins: The paper lacks experimental data on the effects of individual toxins. Including data on the effects of T-2 toxin, deoxynivalenol, and fumonisin B1 separately could help in understanding whether the observed effects are due to a synergistic action or are attributable to individual toxins.
2. Comprehensive Biomarker Analysis: The study should extend beyond mRNA levels of oxidative stress-related genes. It is suggested to measure reactive oxygen species (ROS) and malondialdehyde (MDA) levels in the kidney to provide a more comprehensive analysis of oxidative stress.
3. Protein Expression Levels: In addition to gene expression, assessing the protein expression levels of the studied genes could offer more insight into the actual physiological impact of the toxin mixture on the kidney.
4. Histopathological Evaluation: Including kidney histopathology and renal function markers such as creatinine and uric acid would strengthen the study by providing evidence of morphological and functional changes.
5. Antioxidant Parameters: Measuring antioxidant parameters, such as superoxide dismutase (SOD) and MDA, could enrich the understanding of the antioxidant defense mechanism's response to toxin exposure.
6. Cellular Level Validation: If resources permit, validating the findings at the cellular level could provide foundational support for the observed effects and their mechanisms.
7. A recent study reported that the susceptibilities of DON vary signifcantly among animals, following the order of pigs, mice/rats and poultry from the most to least susceptible (https://doi.org/10.1007/s00204-022-03337-8). This study was conducted in chicken, and do the authors expect that their findings could be varied in other farm animals? I suggest the authors could cite the paper and add a few sentence in discussion to discussion this issue.
Round 2
Reviewer 1 Report
Comments and Suggestions for Authors
Thank you for your responses and clarifications
Reviewer 2 Report
Comments and Suggestions for Authors
no further comment